Fecal fungal microbiota alterations associated with clinical phenotypes in Crohn’s disease in southwest China

Zeng Li 1
Feng Zhe 1
Zhuo Ma 2
Wen Zhonghui 1
Zhu Cairong 3
Tang Chengwei 1
Liu Ling 1 lingzipurple@163.com
Wang Yufang 1 wangyufang04@126.com
1 Department of Gastroenterology, West China Hospital, Sichuan University , Chengdu City, Sichuan Province , China
2 Department of Gastroenterology, Lhasa People’s Hospital , Lhasa, Tibet Autonomous Region , China.
3 School of Public Health and Community Medicine, Sichuan University , Chengdu, Sichuan Province , China
Saki Morteza
Electronic publication date: 2022 Oct 18
Publication date: 2022
Volume: 10
Electronic Location ID: e14260
Received 2022 Jul 27; Accepted 2022 Sep 27
Copyright: © 2022 Zeng et al.
Copyright year: 2022
Copyright holder: Zeng et al.
License: This is an open access article distributed under the terms of the Creative Commons Attribution License, which permits unrestricted use, distribution, reproduction and adaptation in any medium and for any purpose provided that it is properly attributed. For attribution, the original author(s), title, publication source (PeerJ) and either DOI or URL of the article must be cited.
License URL: https://creativecommons.org/licenses/by/4.0/

Keywords: Crohn’s disease, Clinical phenotypes, Fecal fungal microbiota

Funding: National Natural Science Foundation of China 81270447, U1702281, 81670551, 81970463 Sichuan Regional Innovation Cooperation Project 2022YFQ0053 This work was supported by the National Natural Science Foundation of China (81270447, U1702281, 81670551, 81970463), and the Sichuan Regional Innovation Cooperation Project (2022YFQ0053). The funders had no role in study design, data collection and analysis, decision to publish, or preparation of the manuscript.

==============================
Although previous studies reported that gut fungal microbiota was associated with Crohn’s disease (CD), only a few studies have focused on the correlation between gut fungi and clinical phenotypes of CD. Here, we aimed to analyze the association between intestinal fungi and the occurrence of CD, disease activity, biological behaviors, and perianal lesions. Stool samples from subjects meeting the inclusion and exclusion criteria were collected for running internal transcribed spacer 2 (ITS2) high-throughput sequencing. Then, correlation analysis was conducted between intestinal fungi and different clinical groups. There were 45 patients with CD and 17 healthy controls (HCs) enrolled. Results showed that two phyla, Rozellomycota and Mortierellomycota, were not present in patients with CD compared to HCs. At the same time, there was a higher abundance of fungal genera and species belonging to the phylum Ascomycota in patients with CD. SparCC network analysis showed fewer interactions among the fungal communities in patients with CD compared to HCs. Exophiala dermatitidis was positively associated with the clinical active stage and platelet count. The genus Candida was with significantly higher abundance in the non-B1 CD group based on the Montreal classification. Clonostachys, Humicola, and Lophiostoma were significantly enriched in patients with CD with perianal lesions. Our results demonstrated that the composition of the intestinal fungal microbiota in patients with CD and HCs was markedly different, some of which might play a pathogenic role in the occurrence of CD and perianal lesions. Exophiala dermatitidis and genus Candida might be associated with active disease stage and type non-B1 CD (CD with intestinal stenosis or penetrating lesions, or both), respectively.

Introduction

Inflammatory bowel disease (IBD) is a chronic non-self-limited intestinal inflammatory disease that mainly includes Crohn’s disease (CD) and ulcerative colitis (UC). IBD has gradually become a global disease (Ng et al., 2018), so it is of great significance to further study its pathogenesis, diagnosis, and treatment.

While the pathogenesis of IBD is still unclear, an abnormal immune response triggered by environmental factors in individuals with genetic susceptibility is probably related to it (Kostic, Xavier & Gevers, 2014). It is well known that intestinal bacteria are an essential factor in the occurrence and development of IBD (Sartor & Wu, 2017; Li et al., 2018). In addition to intestinal bacteria, intestinal fungi are another important microbe to be studied, especially in patients with CD (Sartor & Wu, 2017).

Human intestinal fungi mainly include Ascomycota, Basidiomycota, and Chytridiomycota at the phylum level (Li et al., 2018), which may interact with the gut mucosal immune system and the occurrence and development of CD. The components of the fungal cell wall mainly include chitin, glucan, and mannan. Among them, the recognition of β-glucan through Dectin-1, a pattern recognition receptor, and activation of the downstream inflammatory pathways have attracted widespread attention in intestinal fungal immunity (Taylor et al., 2007). Furthermore, the anti-Saccharomyces cerevisiae antibody (ASCA) is a highly specific serological marker of CD. Some studies have reported that ASCA may predict the occurrence of CD (Torres et al., 2020) and be related to clinical phenotypes, risk of complications and drug response, etc (Bertha et al., 2017; Duarte-Silva et al., 2019; Kansal et al., 2019; Zholudev et al., 2004).

CD subtypes based on different clinical phenotypes, according to the biological behaviors of Montreal classification (Maaser et al., 2019) may also be associated with diverse intestinal fungi. Although previous studies reported that fungal dysbiosis is related to the occurrence and development of CD (Liguori et al., 2016; Qiu et al., 2020), few studies have documented the correlations of intestinal fungi to the clinical phenotypes of CD. Furthermore, the composition of intestinal fungi was influenced by individual geographical backgrounds, diet, age, and other factors (Clements & Carding, 2018). Therefore, more regional studies are needed to observe the relationship between fungi and clinical manifestations of CD.

Our study aimed to observe the associations between intestinal fungi and disease occurrence, disease activity, biological behavior, and perianal lesions of CD, which may provide helpful information for clinical diagnosis and treatment of CD.

Materials and Methods

Study population

Patients in West China Hospital were enrolled from December 2018 to October 2019. Patients with CD were diagnosed by ECCO-ESGAR Guideline for Diagnostic Assessment in CD (Maaser et al., 2019). Those with infectious diseases were excluded, including gastrointestinal or extra-gastrointestinal infections based on clinical symptoms and serological and radiographic results. In addition, patients with cancer, other gastrointestinal disorders, critical illnesses, or intake of antibiotics and/or probiotics in the last 2 months were excluded. Healthy controls (HCs) were volunteers without any suspicious gastrointestinal discomfort, autoimmune diseases, chronic diseases, or history of any kind of medication within 3 months. All included objects were living in southwest China and were in accordance with a spicy dietary habit. One stool sample was collected from each study participant, keeping at ordinary temperature for less than 10 min before being transferred to an ultra-low temperature freezer (–80 °C) until being sent to the Biozeron (China) for DNA extraction and Illumina PE250 sequencing.

Detailed data, including basic information, clinical manifestation, Crohn’s disease activity index (CDAI), Montreal classification (Satsangi et al., 2006), past history of abdominal surgery, and medication, were collected for further analysis. According to the Montreal classification (Satsangi et al., 2006), the biological behaviors of CD were classified into type B1 (non-stricturing, non-penetrating), type B2 (stricturing), and type B3 (penetrating). As types, B2 and B3 often coexist, the biological behaviors of CD patients were divided into two subgroups in this study, including type B1 and type non-B1 (B2 and/or B3). Patients with CDAI score ≥ 150 were enrolled in the group of CD-flare, or else in the group of CD-remission. Analysis was conducted between intestinal fungi from different groups as follows: 1. CD vs. HC (n1 = 45, n2 = 17); 2. CD-flare vs. CD-remission (n1 = 34, n2 = 11); 3. type B1 vs. type non-B1 (n1 = 20, n2 = 25); 4. CD with perianal lesions vs. CD without perianal lesions (n1 = 25, n2 = 20).

Our ethical approval was obtained from the Biomedical Ethics Committee of West China, Sichuan University (NO. 2016-298). Patients were required to give written consents to the study. For full disclosure, the details of the study are published online on the page of the Biomedical Ethics Committee of West China Hospital, Sichuan University.

DNA extraction and Illumina PE250 sequencing

Microbial DNA was extracted using the HiPure Stool DNA Kits (Magen, Guangzhou, China) according to the manufacturer’s protocols. The fecal samples were homogenized in a lysate and further lysed in a high-temperature water bath, releasing DNA into the lysate. The fecal impurities were removed by centrifugation, and the DNA was further purified through the column. Finally, the DNA was eluted by Buffer AE. After monitoring the concentration and purity of extracted DNA on 1% agarose gels, DNA was diluted to 1 ng/μl using sterile water. The fungal microbiota was identified and analyzed by sequencing the internal transcribed spacer (ITS) fragment, which was the target region ITS2 amplified with universal primers: forward primer (5′-GCATCGATGAAGAACGCAGC-3′) and reverse primer (5′-TCCTCCGCTTATTGATATGC-3′) (Bellemain et al., 2010). We performed PCR in a 30 μl solution with 0.2 μm forward and reverse primers and 10 ng template DNA, which contained 15 μL Phusion® High-Fidelity PCR Master Mix (New England Biolabs, Ipswich, MA, USA). The thermocycling included initial denaturation at 98 °C for 1 min, 30 cycles of denaturation at 98 °C for 10 s, primer annealing at 50 °C for 30 s, extension at 72 °C for 5 min, and final extension at 72 °C for 5 min. For qualification and quantification of our PCR products, we mixed PCR products with the same volume of 1× loading buffer containing SYBR green, performed electrophoresis on a 2% agarose gel, and picked samples with the main band brightness between 400 and 450 bp for further experiments. Eventually, we mixed all obtained PCR products in an isopycnic ratio and refined them with the Gene JET gel extraction kit (Thermo Scientific, Waltham, MA, USA).

The sequencing library was created by the NEB Next® Ultra™ DNA Library Prep Kit for Illumina (NEB, USA), complying with the manufacturer’s instructions, adding index codes. The library was estimated on the Qubit@ 2.0 Fluorometer (Thermo Scientific, Waltham, MA, USA) and Agilent Bioanalyzer 2100 system and sequenced on the Illumina MiSeq platform, finally generating 250/300 bp paired-end reads.

Fungal ITS2 rDNA sequence analysis

Paired-end reads were merged as raw tags using FLASH (version 1.2.11) (Magoč & Salzberg, 2011). Paired-end reads were assigned to each sample based on the unique barcodes. Noisy sequences of raw tags were filtered under specific filtering conditions to obtain high-quality clean tags (Bokulich et al., 2013). The software package UPARSE (version 9.2.64) carried out sequence analyses (Edgar, 2013). Conventionally, operational taxonomic units (OTUs) were defined as sequences with ≥97% similarity. After choosing typical sequences for each OTU, we footnoted category data for them by the RDP classifier (version 2.2) (Wang et al., 2007) based on the ITS2 database (version update_2015), with the confidence threshold value of 0.8 (Ankenbrand et al., 2015).

Statistical analysis

Chao1, ACE, Shannon, Simpson, and Good’s coverage index were calculated in QIIME (version 1.9.1) (Caporaso et al., 2010). PD-whole tree index was calculated in Picante (version 1.8.2) (Kembel et al., 2010). All above demonstrate the α diversity between different groups. Principal coordinate analysis (PCoA) based on weighted and unweighted UniFrac distances was generated in the R project Vegan package (version 2.5.3) demonstrating the β diversity between different groups (Oksanen et al., 2010). Adonis (also called Permanova) and Anosim test was calculated in the R project Vegan package (version 2.5.3) (Oksanen et al., 2010). SparCC network analysis visualized the complex cooperative and competitive relationships of fungi within different groups (Friedman & Alm, 2012). Correlation analysis was calculated in the R project psych package (version 1.8.4), ascertaining the association between intestinal fungi and multiple indicators (Revelle, 2022). Differences with a P-value < 0.05 were considered significant. All authors had access to the study data and reviewed and approved the final manuscript.

Results

Study information and patient characteristics

The information of the 45 enrolled patients is described in Table 1 and Table S1. A total of 17 HCs were recruited during the same period, with an average age of 27.6 ± 9.7 years and a male/female ratio of 8/9. There was no significant difference in sex, mean age, Montreal classification, perianal complications, or surgical history between the CD-flare and CD-remission groups.

Table 1 Study population.

	CD-flare (n = 34)	CD-remission (n = 11)	P value	
Age, Mean ± SD	32.9 ± 11.6	29.4 ± 11.7	0.39	
Sex, n (male/female)	25/9	9/2	0.705	
Montreal classification, n				
A1/A2/A3	1/28/5	1/7/3	0.29	
L1/L2/L3/l4	1/8/25/0	1/3/7/0	0.56	
B1/B2/B3	13/7#/18#	7/3/1	0.07	
Perianal complications, n	20	5	0.50	
Past abdomen surgery history, n	26	6	0.16	
Note:

CD-flare, CD at the active stage (i.e., CDAI score ≥ 150); CD remission, CD at remission stage (i.e., CDAI score < 150); and perianal complications, including perianal fistula and abscess.

Altered intestinal fungal microbiota in patients with CD compared to HCs

At the phylum level, there was no significant difference in the composition of Ascomycota, Basidiomycota, and Mucoromycota between the CD and HC groups (P > 0.05). The phyla Rozellomycota and Mortierellomycota were observed in the HC group, while these two phyla were not present in the CD group (Fig. 1A). The relative abundances of Saccharomyces, Clonostachys, and Exophiala in CD patients were higher than those in HCs, while the relative abundance of Candida was approximately half of that in HCs at the genus level (Fig. 1B). At the species level, the relative abundances of Exophiala dermatitidis, Candida tropicalis, Humicola grisea and some unclassified fungal species were higher in CD patients than in HCs, while the relative abundance of Candida albicans was lower than that in HCs (Fig. 1C) (P < 0.05).

Figure 1 Fecal fungal composition and analysis between CD (n = 45) and HC (n = 17).

Fecal fungal composition at the phylum (A), genus (B) and species (C) levels in CD and HC. The mean relative abundance of fecal fungi from patients with CD and HC at the genus (D, top 15) and species (E, top 10) levels, respectively. Groups were compared using Welch’s t test (all P < 0.05). (F) Beta diversity of fecal fungi from patients with CD and HC. Principal coordinate analysis (PCoA) based on unweighted UniFrac distance with each sample colored according to the disease phenotype. Groups were compared using the PERMANOVA method (Adonis test, P = 0.001). (G) SparCC network analysis showed complex cooperative or competitive intrafungal interactions in the CD group (right) and HC group (left) (all P < 0.05). A larger point size indicates a higher relative abundance; darker colors indicate stronger connectivity; the solid red line indicates a positive correlation; and the broken blue line indicates a negative correlation. See more detailed data about the SparCC network of the CD group in Table S3. Abbreviations: PCoA, principal coordinate analysis; HC, healthy subjects.

Welch’s t-test was used to analyze the mean relative abundance of fecal fungi between these two groups (Fig. 1D). The top 15 genera with significant differences depending on relative abundance were Saccharomyces (P = 0.04), Clonostachys (P < 0.01), Exophiala (P = 0.02), Humicola (P < 0.01), Thermoascus (P < 0.01), Dipodascus (P < 0.01), Cutaneotrichosporon (P < 0.01), Tausonia (P < 0.01), Lophiostoma (P < 0.01), Mucor (P < 0.01), Trichosporon (P < 0.01), Thermomyces (P = 0.04), Cylindrocarpon (P < 0.01), Vanrija (P < 0.01) and Mortierella (P = 0.01). Among them, the top five fungi, depending on relative abundance, all belonged to the phylum Ascomycota. Furthermore, the genera Humicola, Lophiostoma, and Cylindrocarpon only existed in CD patients, all of which were in the phylum Ascomycota. The genus Mortierella was only observed in HC group, with a significant difference compared to CD (P = 0.01). At the species level, there were higher relative abundances of Exophiala dermatitidis (P = 0.02), Humicola grisea (P < 0.01), Thermoascus aurantiacus (P < 0.01), Candida sake, (P < 0.01), Dipodascus australiensis (P < 0.01), Tausonia pullulans (P < 0.01), Cutaneotrichosporon curvatus, (P < 0.01), Mucor racemosus (P < 0.01), Aspergillus sydowii (P < 0.03), and Vanrija fragicola (P = 0.01) in CD patients compared to HCs (Fig. 1E). Among them, the top five fungal species all belonged to the phylum Ascomycota. Humicola grisea, a species belonging to the phylum Ascomycota, was only found in patients with CD. Although two phyla were not present in the CD group, fungi at the genus and species levels belonging to the phylum Ascomycota showed higher relative abundance in the CD group compared with the HC group.

Through the analysis of α diversity indexes, including observed species, Chao1, Shannon, Simpson, and Good’s coverage index, although there was no significant difference between these two groups, patients with CD featured lower fecal fungal richness (observed species, P = 0.27) (Table S2). Principal component analysis (PCoA) based on unweighted UniFrac distance showed the difference in fungal microflora structure between these two groups, indicating that intestinal fungi in some HCs overlapped with those in some CD patients, while some of them were significantly different from the overlapping fungal microflora (Adonis test, P < 0.01) (Fig. 1F). At the genus level, SparCC analysis showed that the number of inter-fungal cooperative or competitive correlations in the CD group was notably reduced compared with that in the HC group (Fig 1G, Table S3). The interacting fungi in the CD group were mainly (7/8) fungi significantly enriched in CD (Figs. 1D and 1G). Among them, the genera Clonostachys, Lophiostoma, and Mucor were positively correlated (Figs. 1D and 1G). The genus Tausonia played a linkage role, was negatively correlated with the above three fungi (absolute value of SparCC correlation > 0.5, P < 0.05), and was positively correlated with the genus Cutaneotrichosporon (SparCC correlation value = 0.51, P < 0.01) (Figs. 1D and 1G).

Fecal fungi in the CD-flare group differed from those in the CD-remission group

The phylum Ascomycota dominated fungi in both CD patients in the active stage (CD-flare, n = 34) and CD patients in the remission stage (CD-remission, n = 11) (Fig. 2A). At the genus level, Candida was proportionable in both the CD-flare group and the CD-remission group. At the same time, Exophiala and Saccharomyces were enriched in the CD-flare group, and Aspergillus and Clonostachys were enriched in the CD-remission group (Fig. 2B). Species Exophiala dermatitidis was also with higher relative abundance in the CD-flare group compared to CD-remission group at the taxonomic levels of its order, family, genus, species, and OTU (P <0.05) (Figs. 2B–2E, Table S4).

Figure 2 Fecal fungal composition and analysis between CD-flare (n = 34) and CD-remission (n = 11).

Fecal fungal composition at the phylum (A), genus (B) and species (C) levels in CD-flare and CD-remission. The mean relative abundance of fungi from CD-flare and CD-remission at the species (D) and OTU (E) levels, respectively. Groups were compared using Welch’s t test (all P < 0.05). The fungal alpha diversity between CD-flare and CD-remission ((F) observed species (P > 0.05); (G) PD-whole tree (P < 0.05)). (H) Beta diversity, altered fungal microbiota biodiversity and composition in CD-flare compared to CD-remission (Adonis test, P = 0.027). (I) Beta diversity, altered fungal microbiota biodiversity and composition in CD-flare, CD-remission, and HC (Adonis test, P = 0.001). Principal coordinate analysis (PCoA) based on unweighted UniFrac distance with each sample colored according to the disease phenotype. Groups were compared using the PERMANOVA method. Abbreviations: PCoA, principal coordinate analysis; HC, healthy subjects; CD-flare, CD at the active stage (i.e., CDAI score ≥ 150); CD remission, CD at remission stage (i.e., CDAI score < 150).

Although no significant difference was observed, the observed species richness of the CD-flare was lower than that of the CD-remission (P = 0.08) (Fig. 2F, Table S5). The PD-whole tree index in CD-flare was significantly lower than that in CD-remission (P = 0.03) (Fig. 2G, Table S5). Principal component analysis (PCoA) based on unweighted UniFrac distance showed a significant difference in β diversity between CD-flare and CD-remission (Adonism test, P = 0.027) (Fig. 2H). Furthermore, there was an apparent separation based on weighted UniFrac distance in the fungal community between CD and a part of HC, while another part of HC overlapped with CD, close to the CD-remission fungal community (Adonism test, P = 0.001) (Fig. 2I). All of the above results suggested a progressive change in the fungal community in HC, CD remission, and CD flare.

Correlation analysis between intestinal fungi and laboratory indicators was conducted, including erythrocyte sedimentation rate (ESR), C-reactive protein (CRP), hematocrit (HCT), white blood cell count (WBC), neutrophil count (N), lymphocyte count (N), neutrophil-to-lymphocyte ratio (N/L), platelet count (PLT), lymphocyte percentage, the ratio of PLT to lymphocyte percent, and lymphocyte percent. The results showed that Exophiala dermatitidis was significantly correlated with increased platelet count at the taxonomic levels of its order (Chaetothyriales, R = 0.34, P = 0.02), family (Herpotrichiellace, R = 0.36, P = 0.02), genus (Exophiala, R = 0.38, P = 0.01) and species (Exophiala dermatitidis, R = 0.37, P = 0.01) (Spearman, all P < 0.05) (Fig. 3A). In addition, at the genus level, a positive correlation was observed between Nigrospora (Spearman, R = 0.36, P = 0.02) and ESR. Verticillium was negatively correlated with ESR (Spearman, R = −0.31, P = 0.04) and CRP (Spearman, R = −0.30, P = 0.04) (Fig. 3A). Lophiostoma, which was enriched in the CD group, was positively correlated with the N/L ratio (Spearman, R = 0.32, P = 0.03) (Figs. 1D and 3A). At the species level, Candida albicans, Verticillium dahliae, Wallemia Canadensis, Aspergillus penicillioides, and Nigrospora oryzae correlated with laboratory indicators to some extent (Spearman, P < 0.05) (Fig. 3B). Exophiala dermatitidis was positively correlated with CDAI and laboratory activity index (platelet count) at multiple taxonomic levels, demonstrating that it may be closely related to CD activity.

Figure 3 Correlation analysis between intestinal fungi and laboratory activity indicators, complications, or extraintestinal manifestations at the genus.

(A) and species (B) levels in patients with CD (Spearman, *P ≤ 0.05; **P ≤ 0.01). Abbreviations: ESR, erythrocyte sedimentation rate; CRP, C-reactive protein; HCT, hematocrit; WBC, white blood cell count; N, neutrophil count; L, lymphocyte count; N/L, neutrophil-to-lymphocyte ratio; PLT, platelet count; PLT lymph percent, the ratio of PLT to lymphocyte percent.

Differed fecal fungi relating to concerned biological behaviors in CD

In terms of complications, at the genus level, Clonostachys (R = 0.34, P = 0.02), Lophiostoma (R = 0.39, P < 0.01), and Fusarium (R = 0.42, P < 0.01) showed positive correlations with fistula formation, abdominal abscess, and arthritis/arthralgia (Spearman, all P < 0.05) (Fig. 3A). Aspergillus (R = −0.39, P < 0.01), Tausonia (R = −0.39, P = 0.04), Trichosporon (R = −0.33, P = 0.02), Wallemia (R = −0.38, P < 0.01), Holtermanniella (R = −0.30, P = 0.04), and Wickerhamomyces (R = −0.31, P = 0.04) were negatively correlated with the formation of fistula, while Vanrija (R = −0.30, P = 0.047) was negatively correlated with the formation of perianal lesions (Spearman, all P < 0.05) (Fig. 3A). The complex relationships between fungi and complications prompted us to pay more attention to the relationships between fungi and biological behavior and perianal lesions in CD.

Notably, type B1 (n = 20) and type non-B1 (B2/B3, n = 25) were concerned biological behaviors in clinical practice. There was no significant difference in baseline parameters between these two groups (P > 0.05) (Table S6). No significant difference was observed in α or β diversity between these two groups. No remarkable difference was found at the phylum level between these two groups (Fig. 4A). At the genus level, the composition proportions of Saccharomyces and Clonostachys were similar between these two groups (Fig. 4B). At the same time, Candida was significantly enriched in type non-B1 CD compared with type B1 (Welch’s t test, P < 0.05) (Figs. 4B and 4D). At the species level, we did not observe a significant difference in Candida albicans, Candida glabrica, or Candida tropicalis in non-B1-type CD (P > 0.05) (Fig. 4C).

Figure 4 Fecal fungal composition and analysis related to different manifestations in patients with CD.

Fecal fungal composition at the phylum (A), genus (B) and species (C) levels in type B1 (n = 20) and type non-B1 (B2+B3, n = 25) based on Montreal classification. (D) The mean relative abundance of fungi from type B1 and type non-B1 at the genus level. Groups were compared using Welch’s t test (P < 0.05). Fungal composition at the phylum (E) and genus (F) levels in patients with perianal lesions (n = 25) or without perianal lesions (n = 20). (G) The mean relative abundance of fungi in patients with or without perianal lesions. Groups were compared using Welch’s t test (all P < 0.05). (H) Beta diversity, altered fungal microbiota biodiversity and composition in patients with or without perianal lesions (Adonis test, P = 0.04). Principal coordinate analysis (PCoA) based on weighted UniFrac distance with each sample colored according to the disease phenotype. PCo1 and PCo2 represent two principal coordinates that capture most of the diversity. The fraction of diversity captured by the coordinate was given as a percentage. Groups were compared using the PERMANOVA method. Abbreviations: PCoA, principal coordinate analysis.

Further analysis was performed grouped by CD patients with perianal lesions (n = 25) and without perianal lesions (n = 20). There was no significant difference in baseline parameters between these two groups (P > 0.05) (Table S7). Fungi at the phylum level were in line with each other (Fig. 4E). The proportion of the genus Candida was similar in these two groups. Clonostachys was enriched in CD patients with perianal lesions, while Saccharomyces and Dipodascus were enriched in CD patients without perianal lesions (Fig. 4F). Furthermore, the fungi significantly enriched in CD patients with perianal lesions at the genus level were Clonostachys (P = 0.02), Humicola (P = 0.02), Lophiostoma (P = 0.03), Fusarium (P = 0.02), Lecanicillium (P < 0.01), and Gibberella (P < 0.01) (Fig. 4G). The relative abundances of these top three fungi, including Clonostachys, Humicola and Lophiostoma, were also significantly higher in CD (Figs. 4G and 1D). No significant difference was observed in α diversity between these two groups, while PCoA based on weighted UniFrac distance showed the different fungal microflora structures between these two groups (Adonism test, P = 0.04) (Fig. 4H).

Discussion

The occurrence of CD involves multiple factors, including different regions with susceptible genetic backgrounds, changes in intestinal microecology, and disorders of the intestinal mucosal immune response (Kostic, Xavier & Gevers, 2014). As the intestinal microbiota is closely related to immunobiology in CD, it has attracted much attention in recent studies. The intestinal microbiota is mainly composed of viruses, bacteria, and fungi. Although the number of fungi is far less than that of bacteria, fungi may play an essential role in the occurrence of CD (Sartor & Wu, 2017).

In this study, we compared the fecal fungal microbiota in patients with CD with HCs. Five fungal phyla were observed in HC; however, Rozellomycota and Mortierellomycota were not present in CD patients. There were some reports of the difference in fecal fungal microbiota between CD patients and HCs; however, the results were quite different from each other (Imai et al., 2019; Li et al., 2014; Liguori et al., 2016; Qiu et al., 2020; Sokol et al., 2017). No research reports that these two fungal phyla were not present in CD patients compared to HCs before, which may be due to different regional environments or dietary habits. Our patients or participants were all from Southwest China, with spicy dietary habit. Bacteria, fungi, and viruses are in dynamic balance in a normal situation. Although we did not sequence and characterize the fecal bacteria in our participants, the inexistence of two fungal phyla in CD patients in our findings may be related to the bacteria alterations, as previous studies suggested the decreased bacterial richness and the increased bacterial abundance in CD (Yilmaz et al., 2019). PCoA demonstrated the difference in fecal fungal microflora structure between CD and HC, indicating that the intestinal fungi in some HCs overlapped with those in some CD patients, while some of them were significantly different from the overlapping fungal microflora. Similarly, Liguori et al. (2016) found consistent characteristics in their study. Fecal fungal genera and species belonging to phylum Ascomycota in CD patients were with significantly higher abundance compared to HCs in our study. Among them, the top five abundance genera and species belonging to phylum Ascomycota were only observed in CD patients, suggesting the phylum Ascomycota may be potentially associated with the occurrence of CD. In our analysis, Candida and Aspergillus were the two most abundant genera, in line with a previous study based on the Chinese population (Qiu et al., 2020). In comparison, Saccharomyces and Debaryomyces are the two most abundant genera based on the western country population (Sokol et al., 2017). Again, the impact of regional differences in the process should be noted.

The relative abundance of the genus Trichosporon was significantly higher in the CD group compared to HC. Previous studies have shown that it is increased in mice with intestinal inflammation and decreased in the noninflammatory mucosa of patients with IBD (Lam et al., 2019). In addition, the genus Trichosporon is potentially associated with other autoimmune diseases. Alonso reported that Trichosporon mucoides was present and/or enriched in nerve tissues of patients with multiple sclerosis (MS), while it was absent in healthy subjects (Alonso et al., 2018). SparCC analysis showed that the fungal interactions at the genus level were reduced in CD patients compared to HCs. The correlated fungi in CD patients were mainly (7/8) enriched in CD patients compared to HCs. It is known that the occurrence of CD is related to the increasing abundance of bacteria with pathogenic effects and the decreasing abundance of bacteria with protective effects. However, there is still a lack of research to support whether fungi have similar characteristics as bacteria above.

In terms of disease activity, the Basidiomycota/Ascomycota ratio was lower in the CD patients compared with HCs, while it increased in the CD-flare compared with CD-remission. Sokol et al. (2017) indicated an increased Basidiomycota/Ascomycetes ratio in IBD patients, especially in CD patients at an active stage, and pointed out there was a higher relative abundance of Basidiomycota in IBD patients. The species Exophiala dermatitidis was closely associated with CD-flare at the taxonomic levels of its order, family, genus, species, and OTU. Exophiala dermatitidis was also significantly positively associated with platelet count, a common important laboratory activity index in clinical practice, at the taxonomic level of its order, family, genus, and species. A recent study reported that Exophiala dermatitidis could be isolated from the oral cavity of HCs (Toberna et al., 2020). However, Exophiala dermatitidis may have pathogenic effects in some cases, as it causes cirrhosis (Hong et al., 2009), diffuse thickening of the common bile duct wall (Oztas et al., 2009), and chronic subcutaneous abscess and granulomatous lesions (Campos-Takaki & Jardim, 1994) in some previous reports. Lemoinne et al. reported an increased abundance of genus Exophiala in patients with primary sclerosing cholangitis (PSC), reminding us that intestinal Exophiala may have some impact on autoimmune disease (Lemoinne et al., 2020). Previous studies have suggested that intestinal microbiota have predictive value in disease activity (Braun et al., 2019). Exophiala dermatitidis is highly associated with the clinical activity and laboratory inflammatory markers of CD. However, the role it plays in CD activity needs further study.

Relevant studies have proposed the predictive value of serological markers in the development of CD; however, the role of fungi in the process has been less studied (Amre et al., 2006; Israeli et al., 2005; Kim et al., 2020). ASCA can potentially predict the development of stenosis and penetrating lesions in the course of CD (Israeli et al., 2005; Ryan et al., 2013; Wu et al., 2019). Based on this, our study further supplements the correlations between fungi and biological behaviors of CD. The genus Candida was significantly enriched in type non-B1 CD, suggesting that Candida may be potentially associated with stenosis and penetrating lesions in CD. It is worth noting that Candida albicans and Saccharomyces cerevisiae have homologous ASCA epitopes, which can be an antigenic stimulation to produce ASCA in different environments, such as under acidic conditions related to its severe pathogenicity (Standaert–Vitse et al., 2006). Candida albicans has attracted attention in many studies of IBD; in addition, some have also pointed out that Candida glabrata (Liguori et al., 2016) and Candida tropicalis (Hoarau et al., 2016) may also be associated with CD. Candida was enriched in CD patients with stenosis and penetrating lesions, which may be related to the predictive value of ASCA to some extent.

Our study only focuses on the correlation between intestinal fungi and CD clinical phenotypes. Recent studies have explored the relationship mechanisms between intestinal fungi and CD. For example, Clostridium innocuum invasion led to intestinal creeping fat (Ha et al., 2020), Debaryomyces Hansenii delayed ulcer healing through the IFN-1-STAT1-CCL5 signaling pathway (Jain et al., 2021), and Malassezia restricted inflammation through the CARD9 pathway (Limon et al., 2019). Further studies are needed to clarify the mechanism of the fungal microbiota in the occurrence and development of CD.

As we enroll subjects with spicy dietary habit in the region of southwest China, matching the baseline of sex, age, and biological behaviors in each between-group comparison, the correlation between intestinal fungi and different CD clinical phenotypes are analyzed in detail under this diet at the first time. Collected samples have been frozen immediately, minimizing alteration due to being kept at room temperature. In addition, there are some limitations to this study. This study is a cross-sectional study that cannot further evaluate the role of fungi in the process of occurrence and development of CD. In addition, it is currently believed that fungi from mucosal specimens are also likely to affect CD, it would be better if we collected mucosal or perianal specimens at the same time. Performing qPCR and thereby getting absolute abundances (Jian, Salonen & Korpela, 2021) combined with relative abundances may be more persuasive. Researches on other kinds of diet are also worthy to be further studied.

Conclusion

In conclusion, the composition of the intestinal fungal microbiota in CD patients and HCs was different, as two fungal phyla were not present in CD patients, and beta diversity analysis demonstrated that CD and HC belong to two diverse fungal gut microbiota. Furthermore, there was a higher relative abundance of fungi belonging to Ascomycota potentially associated with CD compared to HC. There were fewer correlations in fungal correlations, along with fewer interactive restriction between conditional pathogenic fungi in patients with CD compared to HC, which may be related to the occurrence of the disease. Exophiala dermatitidis is positively associated with the clinical active stage and platelet count in CD patients. Some fungi are significantly associated with the formation of fistulas and abdominal abscesses; Candida, an intestinal resident fungus, may be related to intestinal stenosis and penetrating lesions in CD. Fungal genera, including Clonostachys, Humicola, and Lophiostoma, which are significantly enriched in CD, may also play a pathogenic role in the occurrence of perianal lesions.

Supplemental Information

Supplemental Information 1 Supplementary Tables.

Click here for additional data file.

Supplemental Information 2 Detailed information of study population.

Anonymised information of study population including basic information, laboratory test results, Montreal classification, extra-intestinal manifestations, complications and so on.

Click here for additional data file.

We are thankful for the bioinformation analysis support from Gene Denovo Biotechnology Co., Ltd. (Guangzhou, China).

Additional Information and Declarations

Competing Interests

Author Contributions

Human Ethics

Data Availability

The authors declare that they have no competing interests.

Li Zeng performed the experiments, prepared figures and/or tables, authored or reviewed drafts of the article, and approved the final draft.

Zhe Feng performed the experiments, prepared figures and/or tables, authored or reviewed drafts of the article, and approved the final draft.

Ma Zhuo performed the experiments, authored or reviewed drafts of the article, and approved the final draft.

Zhonghui Wen performed the experiments, authored or reviewed drafts of the article, and approved the final draft.

Cairong Zhu analyzed the data, authored or reviewed drafts of the article, and approved the final draft.

Chengwei Tang conceived and designed the experiments, authored or reviewed drafts of the article, and approved the final draft.

Ling Liu conceived and designed the experiments, authored or reviewed drafts of the article, and approved the final draft.

Yufang Wang conceived and designed the experiments, authored or reviewed drafts of the article, and approved the final draft.

The following information was supplied relating to ethical approvals (i.e., approving body and any reference numbers):

The Biomedical Ethics Committee of West China, Sichuan University.

The following information was supplied regarding data availability:

The raw data are available at Sequence Read Archive (SRA) of The National Centre for Biotechnology Information (NCBI): PRJNA861876.

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
