# Peer review of "Fecal fungal microbiota alterations associated with clinical phenotypes in Crohn’s disease in southwest China"

_PeerJ, doi:10.7717/peerj.14260_

## Round 0.1 · original submission · Major Revisions

It is my opinion as the Academic Editor for your article - Fecal fungal microbiota alterations associated with clinical phenotypes in Crohn's disease in southwest China - that it requires a number of Major Revisions.

·

Basic reporting

Here Zeng et al. have investigated the gut mycobiota (or gut fungal microbiota) in Crohn’s disease (CD), and further how the gut mycobiota differ between different clinical phenotypes of CD. The study includes 45 CD patients and 17 healthy controls. Analyses were conducted both between HC and CD and between different phenotypes within the group of CD patients.
The gut mycobiota is scarcely studied, and this study provides new information on the gut mycobiota in CD phenotypes, giving new insights to this topic. Therefore, I find this study suitable for the PeerJ journal, if the study is revised according to the major and minor comments outlined below.

Major comments:
1. Please include introduction to the clinical phenotypes in Chron’s disease since this is now missing from the introduction.
2. Consider performing qPCR and thereby getting absolute abundances instead of relative. Here the relative vs. absolute data is discussed: Jian C, Salonen A, Korpela K. Commentary: How to Count Our Microbes? The Effect of Different Quantitative Microbiome Profiling Approaches. Front Cell Infect Microbiol. 2021 Mar 5;11:627910. doi: 10.3389/fcimb.2021.627910. In addition: in Fig 1 D (and also other places throughout the study): is it mean relative abundance? Relative should be stated in this case. Without absolute abundances and multiple timepoints one cannot state a decrease or increase for example.
3. Please add references to all tools and packages used (no references present in the material and methods part, for example references to FLASH tool, UPARSE, RDP classifier, QIIME, PcoA among others were missing). All packages used in R and tools should be referred to correctly. Similarly, add references to the “statistical analysis” part as well.
4. In the abstract the following is stated: “Stool samples from subjects meeting the inclusion and exclusion criteria were collected for running internal transcribed spacer 2 (ITS2) high-throughput sequencing.”, but in the methods this is stated: “The fungal microbiota was identified and analyzed by sequencing the internal transcribed spacer 96 (ITS) fragment, which was the ITS1 amplifieder of ITS1: ITS1F- ITS2R.”. What does this mean? Was the ITS1 or ITS2 targeted? Additionally, what primers were used? Please add reference! Both of these sentences are confusing and should overall be revised (what is referred to by “ITS1 amplifieder of ITS 1”?).
5. Please include a strengths and limitations part in the discussion, one strength is for example that the samples have been frozen immediately, minimizing alteration due to being kept at room temperature.
6. Please correct the list of references according to the guidelines of the journal!
7. What background factors were used as confounders in the analysis? Please specify these! All relevant background factors should be checked whether these have an impact on the fungal gut composition and if they do, their impact should be eliminated from the analysis. Background factors/confounders are included to decrease the probability of the results stemming from the background factors, and not from the actual comparison.

Minor comments:
1. References: in the instructions for authors of the PeerJ journal it is stated: “PeerJ uses the 'Name. Year' style with an alphabetized reference list.”, please modify accordingly.
2. Line 55: please correct nonself to non-self. Further, non-self-limited intestinal inflammatory disease is somewhat confusing and thus could be clarified.
3. Lines 56-58: “In addition to intestinal bacteria, intestinal fungi are another important microbe to be studied, especially in patients with CD.” why particularly in patients with CD?
4. Line 88: please be more specific on the collection; did you collect one stool sample from each study participant?
5. Line 93: please add reference or description to the CTAB/SDS method used in DNA extraction.
6. Line 103: what further experiments? Would there be a problem choosing samples that only included specific bands? The product from ITS PCR can vary quite heavily in size (200-600 bp) (Callahan BJ, McMurdie PJ, Rosen MJ, Han AW, Johnson AJ, Holmes SP. DADA2: high-resolution sample inference from Illumina amplicon data. Nat Methods 2016;13:581–3.).
7. Fungal ITS1-2 rDNA Sequence Analysis part of materials and methods: line 111: Pair-end reads… was merged?
8. Line 114: what database was used for annotation – please specify database and version and add this to the list of references.
9. Line 147: disappeared would mean that the two phyla were present in the CD group first but disappeared over time. Please specify to something in style with “was not present”.
10. Line 156: Saccharomyces  Saccharomyces
11. Line 167: compared to HC?
12. Line 175: should it be stated in methods?
13. Line 231- The Montreal classifications might be more suitable to be explained in methods than in results
14. Line 245: compared to Saccharomyces, and Dipodascus that were enriched..
15. Line 249: the relative abundances..
16. Line 262: disappeared word used again
17. Line 264: Different how? Different from the results of this study? Or Different from each other? Please specify.
18. Line 265: disappeared: does it mean did not exist or did you have healthy controls who later developed CD and thereby it disappeared?
19. Line 268: did you also characterize the bacteria? Did they have decreased richness and abundance? And did you characterize the absolute abundance? If not, then increased abundance cannot be stated, since that can be characterized only by absolute abundance.
20. Line 271: only last name in an in-sentence citation
21. Line 273: please refer to HC throughout the study
22. Line 297-289: Please revise the sentence: “It is known that the occurrence of CD is associated with increasing bacteria with stronger pathogenic effects and decreasing bacteria with protective effects”, it is difficult to understand.
23. Line 290: Please revise the sentence: “However, there is still a lack of research to support whether fungi have similar characteristics as bacteria as above, as well as mutually cooperative and competitive interactions in fungi, viruses, and bacteria.”, I don’t understand what the message is: what does “as bacteria as above” mean? And is the latter part of the sentence also under the “lack of research”? Please clarify, it is very confusing!
24. Line 294 and 295: results should be in the results part of the article. In the discussion the results are discussed, not stated, nor repeated. Additionally, consider removing the “as” in line 292, it might not be suitable.
25. Line 295: Maybe indicated instead of proved? It is not a proof-of-concept study
26. Line 297: increase in the relative ? abundance of Basidiomycota in what group? compared to what? Or was it a higher relative abundance? An increase is that it has been lower and is now higher.
27. Line 298: again, please include numerical results in the results (referring to the score)
28. Line 305: should Exophiala be in italics? Maybe include what taxonomical level to avoid confusion
29. Line 335: disappeared or was not present?
30. Line 336: They  CD and HC belong to two diverse fungal gut microbiota
31. Line 336: increased or higher? Increased means that it has been lower and is now higher, and to this, absolute abundance and multiple timepoints is required. If higher then the comparison has to be stated: there was a higher relative abundance of fungi belonging to Ascomycota … in WHAT GROUP compared to ?
32. Line 337: decrease or fewer correlations? and 338: reduction?
33. Data availability statement: The raw data have been submitted – Is raw data = sequences? Please specify.
34. Figure 1 C, E, 2 C and 3B: please modify the taxa names: Candida_tropicalis  Candida tropicalis
35. Figure 1: Please specify: gut/fecal fungal microbiota composition in patients with Crohn’s disease (CD) (n =45) and healthy controls (HC) (n = 17). Might be clearer to open up abbreviations in the text compared to adding it separately in the end.
36. Further: Add gut/fecal microbiota composition. Fungal composition can be from any body site.
37. Figure 2: Add gut/fecal fungal microbiota composition to clarify
38. The quality of the figures is poor, but I am assuming that high-quality figures will be attached in the final submission according to the guidelines of the journal.

Experimental design

All comments are listed in the 1st section

Validity of the findings

All comments are listed in the first section

·

Basic reporting

no comment

Experimental design

no comment

Validity of the findings

In this study, The whole manuscript describes the association between intestinal fungi and the occurrence of CD, disease activity, biological behaviors, and perianal lesions. The results showed a disappearance of two phyla, Rozellomycota and Mortierellomycota, and an increased abundance of fungal genera and species belonging to the phylum Ascomycota in CD patients. Exophiala dermatitidis and genus Candida might be associated with active disease stage and type non-B1 CD respectively. These results provide some evidences for the study of the mechanism of IBD. Some suggestions are listed below, hope it will be helpful to improve this manuscript.

Additional comments

L56. intestinal bacteria
L57. intestinal fungi is
L57. "especially in patients with" Intestinal fungi not only play a role in CD but also UC. For example, previous studies reported that that UC patients, accompanying with the alteration of fungal biodiversity and composition, were heavily colonized with C. albicans, which aggravated mucosal injury and generation of anti-S. cerevisiae antibodies (ASCA) (Ott, 2008; Colombel, 2013).
L60. immune system
L69.intestinal fungi is
fungi are commonly found on food product or part of food products production (for flavour and aroma) so they can become a continuous source of fungi. In this study, fungal results were very interesting. Wether this study examine the dietary habits of CD patients?
Anti-Saccharomyces cerevisiae antibody (ASCA) is a highly specific serological marker of CD. whether the level of ASCA in subjects and CD patients was detected?
Sovran et al. 2018. found that E. coli treatment allowed the colonization of S. boulardii and C. albicans and restored both beneficial effects of S. boulardii and pathogenic effects of C. albicans on colitis severity in the mice treated with specific antibiotic. whether the sequencing analysis of bacterial microbiota in subjects and CD patients was performed?
In Figure 4. Stool samples were performed to study the relationship between the fungal microflora and biological behaviors. Whether you do the fungal analysis from perianal sample?
L263. There were
L263. some reports of
L264-266. "No research reports that these two fungal phyla disappeared in CD patients compared to HCs......" Please rewrite it.
L272. their study
L273-276. "Among them, both the top five fungi depending on relative abundance and fungi only observed in CD patients belong to the......" please rewrite it.
L282-283. "In addition, the genus Trichosporon is potentially associated with other autoimmune diseases." please add the reference.

---

## Round 0.2 · accepted · Accept

Dear Dr. Wang,

Thank you for your submission to PeerJ.

I am writing to inform you that your manuscript - Fecal fungal microbiota alterations associated with clinical phenotypes in Crohn's disease in southwest China - has been Accepted for publication.

·

Basic reporting

The revised version of the manuscript was thoroughly corrected, I only have minor comments:
Line 43: The genus Candida was significantly more abundant in .. OR The non-B1 CD group based on the Montreal classification had a significantly higher abundance of more
Line 36: differed markedly
Line 59: use only one type of citation
Line 313: Sokol et al. not H
Line 340: The citations should not be in italics

Experimental design

-

Validity of the findings

-

Additional comments

-

·

Basic reporting

no comment

Experimental design

no comment

Validity of the findings

no comment

Additional comments

The authors present results that provide foundation for the further study of the role of fungal microbiota in CD. In my opinion, this manuscript should be accepted by the PeerJ.